Altered proteome of a Burkholderia pseudomallei mutant defective in short-chain dehydrogenase affects cell adhesion, biofilm formation and heat stress tolerance

Reamtong Onrapak 1
Indrawattana Nitaya 2
Rungruengkitkun Amporn 2
Thiangtrongjit Tipparat 1
Duangurai Taksaon 2 3
Chongsa-nguan Manas 2
Pumirat Pornpan pornpan.pum@mahidol.ac.th 2
1 Department of Molecular Tropical Medicine and Genetics/Faculty of Tropical Medicine, Mahidol University , Bangkok , Thailand
2 Department of Microbiology and Immunology/Faculty of Tropical Medicine, Mahidol University , Bangkok , Thailand
3 Department of Companion Animal Clinical Sciences/Faculty of Veterinary Medicine, Kasetsart University , Bangkok , Thailand
Pacheco Luis
Electronic publication date: 2020 Mar 19
Publication date: 2020
Volume: 8
Electronic Location ID: e8659
Received 2019 Nov 8; Accepted 2020 Jan 29
Copyright: ©2020 Reamtong et al.
Copyright year: 2020
Copyright holder: Reamtong et al.
License: This is an open access article distributed under the terms of the Creative Commons Attribution License, which permits unrestricted use, distribution, reproduction and adaptation in any medium and for any purpose provided that it is properly attributed. For attribution, the original author(s), title, publication source (PeerJ) and either DOI or URL of the article must be cited.
License URL: https://creativecommons.org/licenses/by/4.0/

Keywords: Burkholderia pseudomallei, Short-chain dehydrogenase, Proteome, Adhesion, Biofilm formation, Heat stress

Funding: Mahidol University, fiscal years 2016 and 2018 This work was supported by research grants by Mahidol University, fiscal years 2016 and 2018. The funders had no role in study design, data collection and analysis, decision to publish, or preparation of the manuscript.

==============================
Burkholderia pseudomallei is a Gram-negative bacillus that causes melioidosis and is recognized as an important public health problem in southeast Asia and northeast Australia. The treatment of B. pseudomallei infection is hampered by resistance to a wide range of antimicrobial agents and no vaccine is currently available. At present, the underlying mechanisms of B. pseudomallei pathogenesis are poorly understood. In our previous study, we reported that a B. pseudomallei short-chain dehydrogenase (SDO; BPSS2242) mutant constructed by deletion mutagenesis showed reduced B. pseudomallei invasion and initial intracellular survival. This indicated that SDO is associated with the pathogenesis of melioidosis. In the present study, the role of B. pseudomallei SDO was further investigated using the SDO deletion mutant by a proteomic approach. The protein profiles of the SDO mutant and wild-type K96243 were investigated through gel-based proteomic analysis. Quantitative intensity analysis of three individual cultures of the B. pseudomallei SDO mutant revealed significant down-regulation of five protein spots compared with the wild-type. Q-TOF MS/MS identified the protein spots as a glutamate/aspartate ABC transporter, prolyl-tRNA synthetase, Hsp70 family protein, quinone oxidoreductase and a putative carboxypeptidase. Functional assays were performed to investigate the role of these differentially expressed proteins in adhesion to host cells, biofilm induction and survival under heat stress conditions. The SDO deletion mutant showed a decreased ability to adhere to host cells. Moreover, biofilm formation and the survival rate of bacteria under heat stress conditions were also reduced in the mutant strain. Our findings provide insight into the role of SDO in the survival and pathogenesis of B. pseudomallei at the molecular level, which may be applied to the prevention and control of B. pseudomallei infection.

Introduction

Burkholderia pseudomallei is the etiological agent of melioidosis in both humans and animals. It is a natural inhabitant of soil, stagnant water and rice paddies, where the disease is endemic. Melioidosis endemic areas include South-East Asia, in particular northeastern Thailand and northern Australia (Wuthiekanun et al., 1995; Cheng & Currie, 2005). Rice farmers are considered to be at high risk of exposure during the monsoonal and rainy season (Chaowagul et al., 1989; Cheng & Currie, 2005; Inglis & Sagripanti, 2006; Wiersinga et al., 2006), particularly when planting and harvesting in mud and surface water in rice fields. Infection occurs by inoculation through skin abrasions or inhalation, and ranges from acute to chronic. Acute infection often involves septicemia, resulting in death within days of exposure. The longest reported incubation period between initial acquisition and subsequent infection is a remarkable 62 years. Furthermore, a high rate of relapse has been recognized (Ngauy et al., 2005). B. pseudomallei exhibits resistance to diverse groups of antibiotics including third-generation cephalosporins, penicillins, rifamycins, macrolides, quinolones and aminoglycosides (Cheng & Currie, 2005). The antibiotic ceftazidime is the drug of choice for treatment, either alone or in combination with other antibiotics such as chloramphenicol, doxycycline and trimethoprim-sulfa methoxazole. Currently, there is no vaccine available against melioidosis. Owing to its aerosol infectivity, the severe course of infection, and the absence of vaccines and fully effective treatments, B. pseudomallei is classified as a hazard category three pathogen and considered a potential biothreat agent (Cheng & Currie, 2005; Cheng, Dance & Currie, 2005).

B. pseudomallei can survive as a free-living organism in environmental niches, such as soil and water; as well as being parasitic to living organisms such as amoeba, plants, fungi and animals (Inglis & Sagripanti, 2006). Furthermore, this bacterium is able to live and multiply under various adverse conditions (Choy et al., 2000; Dance, 2000). It is likely that this organism has developed strategies to survive in both the natural environment and in its respective hosts. In the human host, B. pseudomallei can infect both phagocytic and non-phagocytic cells. Once the bacteria adheres to the target receptor, it invades the host cell cytoplasm. Following internalization, B. pseudomallei has evolved mechanisms to enter and escape from phagosomes, passing into the host cell cytosol (Wiersinga et al., 2006). In addition, B. pseudomallei induces actin polymerization that leads to bacterial motility and the formation of host-cell-membrane protrusions tipped by intracellular bacteria that project into adjacent cells (Stevens et al., 2005). These protrusions are believed to underlie the ability of B. pseudomallei to uniquely promote multinucleated giant cell formation (Kespichayawattana et al., 2000). This characteristic phenotype has been observed in the tissues of melioidosis patients (Wong, Puthucheary & Vadivelu, 1995). However, the molecular mechanisms behind this pathogen’s survival and pathogenesis remain largely unknown.

The short-chain dehydrogenase/oxidoreductase (SDO) is an enzyme that catalyzes reversible NAD(P)(H)-dependent reactions (Kallberg, Oppermann & Persson, 2010). In the dehydrogenase reaction, a hydride (i.e., a proton plus two electrons) is removed from the substrate and transferred to an electron acceptor, which depending on the enzyme is NAD + or NADP +. The role of SDO in bacterial pathogenesis has been described in Pseudomonas aeruginosa. P. aeruginosa SDO is involved in reduced pyocyanin production, decreased motility, poor biofilm formation and absent paralysis of Caenorhabditis elegans (Bijtenhoorn et al., 2011). Recently, SDO-defective B. pseudomallei showed a significant reduction in glucose dehydrogenase (GDH) activity, invasion and survival in host cells (Pumirat et al., 2014). However, the underlying mechanism of SDO in the pathogenesis of this bacterium remained unknown.

In the present study, we explored the roles of SDO in the pathogenesis and survival of B. pseudomallei. We performed proteomic analysis to investigate SDO-associated proteins in B. pseudomallei. The cellular proteome of K96243 wild-type was compared with that of the SDO mutant using a gel-based approach. The phenotypes associated with the differentially expressed proteins were studied in the wild-type and mutant strains, and included adherence to host cells, biofilm formation and survival under heat stress. Taken together, our data provide insight into the molecular mechanisms of SDO in B. pseudomallei pathogenesis and survival.

Materials and Methods

Bacterial strains, cell lines and culture conditions

B. pseudomallei wild-type K96243, the SDO mutant (Pumirat et al., 2014) and the complemented strain (Pumirat et al., 2014) were cultured in Luria-Bertani (LB) medium (Difco™, Becton Dickinson, USA) and grown at 37 °C. To determine the growth kinetics of B. pseudomallei, the overnight culture of B. pseudomallei adjusted to the optical density at 600 nm (OD600) 0.5 was inoculated 1:500 into standard LB broth. Every 2 h after being inoculated, the optical density of cultures at various time points was recorded.

Cell lines used in this study included A549 (human respiratory epithelial cells) and HFF-1 (human skin fibroblast), which were obtained from the American Type Culture Collection (ATCC, Manassas, VA, USA). The A549 cell line was maintained in Ham’s F-12 medium supplemented with 10% (v/v) heat-inactivated fetal bovine serum (FBS), while the HFF-1 skin fibroblast cell line was maintained in Dulbecco’s modified Eagle’s medium (DMEM) supplemented with 10% (v/v) heat-inactivated FBS. All cells were cultured in a 5% CO2 atmosphere at 37 °C in a humidified incubator.

Protein lysate preparation

B. pseudomallei wild-type (K96243) and SDO mutant cells were resuspended in lysis buffer containing of 8 M urea (OmniPur®, Germany), 2 M thiourea (Merck, Germany), 4% CHAPS (Thermo Scientific, USA) and 50 mM dithiothreitol (DTT) (OmniPur®). The lysates were ultrasonicated on ice and the supernatants were collected after centrifugation at 12, 000 × g for 5 min at 4 °C. A 2-D clean-up kit (GE Healthcare, Germany) and Quick Start Bradford protein assay (Bio-Rad, USA) were used for protein precipitation and quantification.

Two-dimensional gel electrophoresis (2-DE)

Proteins were rehydrated on a 7-cm immobilized pH gradient (IPG) strip (pH 3–10, NL) (GE Healthcare) overnight in 5 M urea, 2 M thiourea, 50 mM DTT, 4% CHAPS and IPG buffer. After isoelectric focusing by an Ettan™ IPGphor™ 3 (GE Healthcare), proteins were reduced in 50 mM DTT for 15 min and alkylated in 125 mM iodoacetamide (IAA) in 6 M urea, 75 mM Tris-HCl, 70 mM SDS and 30% glycerol for 15 min. The samples were further separated by 12% acrylamide gel (Bio-Rad). Following 2-DE, all gels were stained with Coomassie blue. Three biological replicates were performed for each sample. The spot volume was used for quantification. Spots of interest that showed at least a two-fold difference and an ANOVA P value ≤ 0.05 were excised for protein identification.

In-gel tryptic digestion

Each gel piece was destained with 50% acetonitrile (ACN) in 50 mM ammonium bicarbonate (Merck, USA). Proteins in gel spots were reduced and alkylated by 4 mM DTT and 250 mM IAA, respectively. The samples were dehydrated with 100% ACN (Thermo Scientific, USA). Trypsin (Sigma-Aldrich, USA, T6567) was added for digestion overnight at 37 °C. After extraction with 100% ACN, peptides were stored at −20 °C.

Mass spectrometry analysis

An Ultimate® 3000 Nano-LC system (Thermo Scientific) was used for peptide separation. A microTOF-Q II (Bruker, Germany) was used to analyze MS and MS/MS spectra at m/z 400–2,000 and m/z 50–1,500, respectively. The acquisition was controlled by HyStar™ version 3.2 (Bruker). DataAnalysis™ software version 3.4 (Bruker) was used to convert raw data format (.d) files to mascot generic files (.mgf), which were further searched by Mascot software (Matrix Science, USA). A SwissProt bacterial database was set for protein identification.

RNA preparation and real-time RT-PCR

RNA was isolated from stationary phase growth of B. pseudomallei cells grown at 37 °C by adding 10 ml of RNAprotect bacterial reagent (Qiagen) to 5 ml of bacterial culture and incubating for 5 min at room temperature. Subsequently, total RNA was extracted from the bacterial pellets using Trizol (Invitrogen, Carlsbad, CA, USA) according to the manufacturer’s instructions and was treated with DNase (NEB, MA, USA) for 10 min at 37 °C before use. Standard PCR for the 23S RNA gene was used to verify that there was no gDNA contamination in the DNase-treated RNA samples. Real time RT-PCR was performed using the Brilliant II SYBR® Green QPCR Master Mix, one step (Agilent Technologies, Santa Clara, CA, USA). Amplifications of five genes (kex, wzt, qor, proS and hsp) were performed under the following conditions: reverse transcription at 50 ° C for 30 min, enzyme activation at 95 °C for 10 min, then 40 cycles of denaturation at 95 °C for 30 s, annealing at 55 °C for 1 min, and melting curve analysis at 72 °C for 1 min in a CFX96 Touch™ Real-Time PCR Detection System (Bio-Rad, Singapore) as previously described (Pumirat et al., 2017). The primer sequences are shown in Table 1. Relative mRNA levels were determined by fold-changes in expression and calculated by 2−ΔΔCT using the relative mRNA levels of 23S RNA, and the expression of a representative house-keeping gene as a baseline for comparison.

Table 1 Primers used in this study.

Primer name	Primer sequence (5′–3′)	Product size (bp)	Encodedprotein	Source	
Kex F	GTCGAGAATCGACGACTG	150	Putative carboxypeptidase	This study	
Kex R	CGCTATCTGACGAAGCAC	
Wzt F	GACGACCTGCTGATTCTG	191	Glutamate/aspartate ABC transporter	
Wzt R	CCAAGGAGATGACAACGA	
Qor F	CACGTCCGCTTACCTGAT	154	Quinone oxidoreductase	
Qor R	CTTCTCGTCGCTCGACAC	
ProS F	AGATGCCGGTGAACTTCT	165	Prolyl-tRNA synthetase	
ProS R	CGTACGCGTCGTACATCT	
Hsp F	GGCGAACATATTCTGCTG	176	Hsp70 family protein	
Hsp R	GGAACTGCTTGTGCTGAC	
23s F	TTTCCCGCTTAGATGCTTT	343	23S RNA	Pumirat et al. (2010)	
23s R	AAAGGTACTCTGGGGATAA	

Heat resistance assay

A heat stress resistance assay was performed as described previously (Pumirat et al., 2017) with some modifications. Briefly, B. pseudomallei cultured in LB medium at 37 °C for 6 h were washed with phosphate-buffered saline (PBS) and resuspended in PBS to an OD600 of 0.15. One milliliter of the bacterial suspension was then added into a prewarmed tube and incubated at 50  °C for 15 min. Before and after heat challenge, bacterial survival was enumerated on LB agar plates after incubating at 37 °C for 24 h. The number of surviving bacteria was expressed as a percentage of the viable cells. % Survival = CFU (heat exposure) ×100/CFU (without heat exposure)

Biofilm formation assay

Analysis of B. pseudomallei biofilm formation was performed by a microtiter-plate assay as previously described (Pumirat et al., 2017). Each B. pseudomallei strain was assayed with at least eight replicates per experiment, along with positive and negative controls. The optical density was measured at 630 nm using a microplate reader (Bio-Rad). The biofilm formation capacity was calculated as the OD630 of the test strain divided by the OD630 of the negative control.

Adhesion assay

The adhesion of B. pseudomallei with human respiratory and skin cell lines was studied as previously described (Essex-Lopresti et al., 2005). Bacterial inocula were prepared from overnight cultures grown in LB broth, incubated statically for 18 h at 37  °C. Monolayers were infected with diluted bacterial cultures at a multiplicity of infection (MOI) of 100 for 1 h at 37 °C. Non-adherent bacteria were removed by five washes with PBS. Monolayers were lysed with 0.1% (vol/vol) Triton X-100 for 30 min at 37 °C, and adherent cell-associated bacteria were enumerated by plate counts.

Statistical analysis

All assays were conducted at least in triplicate and statistical analyses of independent experiments were performed by ANOVA with a 5% confidence interval in GraphPad Prism 5 program (Statcon). Results were considered significant at a P value of <0.05.

Results

Altered proteome in the B. pseudomallei SDO mutant

As SDO activity facilitates B. pseudomallei invasion and affects the initiation of successful intracellular infection, we predicted that the expression of several proteins may be modulated by SDO. To investigate the molecular mechanisms of SDO, we performed proteomic analysis of the SDO mutant compared with the K96243 wild-type. Proteins extracted from wild-type and mutant bacteria were resolved in 2-D gels, individually (n = 3 gels for each group; total n = 6 gels). Figure 1 shows representative 2-D gels of cellular proteins extracted from the K96243 wild-type and SDO mutant. Up to 497 protein spots were visualized by Coomassie blue staining. Among these, quantitative intensity analysis and statistics revealed five differentially expressed protein spots (with P < 0.05) between the K96243 wild-type and SDO mutant strains. These differentially expressed proteins were subsequently identified by LC-MS/MS analysis. The protein identification results, as well as the quantitative data, are shown in Table 2 and Supplemental Information 1–5. The five differentially expressed proteins were identified as a glutamate/aspartate ABC transporter, prolyl-tRNA synthetase, Hsp70 family protein, quinone oxidoreductase and putative carboxypeptidase.

Figure 1 Proteomic profiles of Burkholderia pseudomallei wild-type K96243 and the SDO mutant.

Table 2 Altered protein expression between B. pseudomallei wild-type K96243 and the SDO mutant.

Protein	Accession no.	M.W.	pI	%Cov	No. of peptide	K96243	SDO	Ratio	ANOVA	
Putative carboxypeptidase	CDU31600	60112	5.5	43.9	19	0.31	0.05	0.15	0.009	
Glutamate/aspartate ABC transporter	ABN82061	36306	7.9	48.8	15	0.15	0.02	0.14	0.045	
Prolyl-tRNA synthetase	SYP_BURMA	63413	5.5	28.4	14	0.19	0.04	0.23	0.035	
Hsp70 family protein	WP_004524146	65225	5.7	13.8	5	0.1	0	0	0.006	
Quinone oxidoreductase	WP_027716432	35439	5.5	2.4	1	0.06	0	0	0.001	

To confirm the level of gene expression, these five proteins were quantified by qRT-PCR. In comparison to the wild-type strain, the expression levels of these five genes were obviously decreased in the SDO mutant (Fig. 2), which was consistent with the proteomic findings. The SDO complement strain was able to recover the expression levels of these five genes (Fig. 2).

Figure 2 Fold changes in gene expression of the kex, wzt, qor, proS and hsp genes in Burkholderia pseudomallei.

Besides, the growth of B. pseudomallei wild-type, SDO mutant and SDO complement strain was determined to exclude the possibility that protein and RNA expression levels may result from the difference of bacterial growth. No significant difference in growth among B. pseudomallei strains (Supplemental Information 6).

Reduced adhesion of the B. pseudomallei SDO mutant to host cells

The proteomic analysis results revealed that the glutamate/aspartate ABC transporter and Hsp70 family proteins were down-regulated in the SDO mutant. These two proteins have been reported to play a role in bacterial adhesion (Leon-Kempis Mdel et al., 2006; Ghazaei, 2017). Adhesion is important for bacterial survival and the spread of bacterial cells, and is therefore a phenotype associated with virulence. B. pseudomallei is a facultative, intracellular bacteria that is able to adhere to, and invade, host cells (Kespichayawattana et al., 2000). Hence, we investigated the involvement of SDO in B. pseudomallei adhesion. B. pseudomallei wild-type and the SDO mutant were examined for their ability to adhere to A549 human respiratory and HFF-1 human skin fibroblast cell lines; since infection with this bacterium occurs by inoculation through inhalation and skin abrasions. As shown in Fig. 3A, the B. pseudomallei SDO mutant showed a significantly lower level of adherence compared with the K96243 wild-type to the A549 cell line (P = 0.0026) and to the HFF-1 cell line (P = 0.0205). The SDO complemented strain recovered the adhesion of the SDO mutant to a similar level to that of the wild-type. These data suggested the role of SDO in the in vitro adherence of B. pseudomallei to the A549 cell line.

Figure 3 Phenotypic examination of Burkholderia pseudomallei.

Decreased biofilm formation in the B. pseudomallei SDO mutant

Down-regulation of the Hsp70 family protein in the SDO mutant has been reported to be involved in biofilm formation (Arita-Morioka et al., 2015). The ability of bacteria to form biofilm is crucial for their survival in adverse environments. Some dehydrogenases are necessary for biofilm formation by bacteria (Bijtenhoorn et al., 2011). To investigate the role of SDO in biofilm formation, the abilities of B. pseudomallei wild-type and the SDO mutant to induce biofilm were evaluated (Fig. 3B). The SDO mutant showed a significantly reduced ability to induce biofilm compared with the wild-type strain (P = 0.0065), suggesting that SDO plays a role in the biofilm formation of B. pseudomallei.

Impaired heat stress tolerance of the B. pseudomallei SDO mutant

Many studies have reported that dehydrogenases are associated with protection of bacterial cells against environmental stress (Fu, Hassett & Cohen, 1989; Cabiscol, Tamarit & Ros, 2000; Liu et al., 2001; Messner & Imlay, 2002; Hoper, Volker & Hecker, 2005; Weerakoon et al., 2009; Miller et al., 2010). We also found that the SDO mutant down-regulated the expression of Hsp70 family protein and quinone oxidoreductase, which plays a role in heat stress tolerance (Liu et al., 2008; Ghazaei, 2017). Thus, the effect of SDO on heat resistance in B. pseudomallei was evaluated. B. pseudomallei wild-type and the SDO mutant were cultured in LB broth, followed by heating at 50 °C for 15 min. As shown in Fig. 3C, a significant difference in heat resistance was detected between B. pseudomallei wild-type and the SDO mutant (P = 0.0343). The mean and standard deviation (SD) of bacterial survival in medium containing 150 mM of B. pseudomallei wild-type after heat treatment were 12.7 ± 2.5%. By contrast, the mean and SD of bacterial survival of the SDO mutant were 8.2 ± 1.0%. These data clearly revealed that SDO was associated with the resistance of B. pseudomallei to heat stress.

Discussion

B. pseudomallei is a soil saprophyte and the causative agent of melioidosis, a disease endemic in southeast Asia and northern Australia (Wuthiekanun et al., 1995). This bacterium can survive for prolonged periods under various environmental conditions and in various hosts. B. pseudomallei possesses a variety of bacterial factors/enzymes to facilitate its survival and pathogenesis. Among these, SDO is an enzyme that plays a role in the pathogenesis of several bacteria including P. aeruginosa (Bijtenhoorn et al., 2011) and B. pseudomallei (Pumirat et al., 2014). In B. pseudomallei, SDO is essential for the processes of invasion and intracellular replication (Pumirat et al., 2014). This suggests that SDO activity may modulate the expression of several essential proteins to facilitate the successful infection of B. pseudomallei.

To obtain an overall view of the SDO-modulated response of B. pseudomallei, we performed a comparative proteomic analysis using the 2-DE technique and mass spectrometry. Interestingly, five differentially expressed proteins were identified, namely, a glutamate/aspartate ABC transporter, prolyl-tRNA synthetase, Hsp70 family protein, quinone oxidoreductase and a putative carboxypeptidase. All of these proteins were down-regulated (SDO mutant/K96243 wild-type ratios ranged from 0.00 to 0.23; average = 0.10). Although only a limited set of differentially expressed proteins could be detected between the SDO mutant and K96243 wild-type, these proteins were able to provide a starting point for more detailed analysis of SDO-modulated bacterial pathways.

In general, the bacterial response to environmental stress is orchestrated by the expression of a family of proteins termed the heat shock proteins, which include Hsp70 family protein. Hsp70 protein is essential for bacterial growth under various stress conditions including high temperature (Ghazaei, 2017). Heat shock proteins play a significant role in maintaining correct protein configurations to avoid cellular damage. Furthermore, Hsp70 protein plays a role in pathogenesis (Ghazaei, 2017) and biofilm formation (Arita-Morioka et al., 2015). During infection, bacteria activate their heat shock genes to protect their cellular machinery from host defense mechanisms, thereby enhancing their virulence. Therefore, down-regulation of Hsp70 family protein in the SDO mutant was concordant with the phenotype of the SDO mutant, which showed impaired heat stress tolerance, biofilm formation and adhesion ability.

Carboxypeptidase is a class of enzymes that hydrolyzes peptides, dipeptides or longer homologs from the C-terminal. The activity of this enzyme has been detected in a number of clinical pathogens including Acinetobacter baumannii, Campylobacter jejuni, Listeria monocytogenes, Pseudomonas aeruginosa and Streptococcus agalactiae (Lough et al., 2016). A report showed that the hydrolysis of peptidoglycan LD-carboxypeptidase Pgp2 influences the helical cell shape and pathogenic properties of C. jejuni (Frirdich et al., 2014). Thus, it is possible that carboxypeptidase plays a synergistic role with SDO in the hydrolysis of the protein substrates for B. pseudomallei infection.

Bacterial ATP binding cassette (ABC) transporters function as versatile systems for the import and export of a variety of molecules across cell membranes (Holland & Blight, 1999). The significance of the glutamate/aspartate ABC transporter in bacterial pathogenesis has been reported in C. jejuni. The PEB1 aspartate/glutamate ABC transporter serves as an adhesin for C. jejuni adhesion (Leon-Kempis Mdel et al., 2006). Here, expression of the glutamate/aspartate ABC transporter protein was found to be reduced in the SDO mutant, leading to defective adhesion. This suggested that the glutamate/aspartate ABC transporter protein was involved in adhesion of the B. pseudomallei SDO mutant.

Quinone oxidoreductase is a multisubunit integral membrane enzyme that operates in the respiratory chains of both bacteria and eukaryotic organelles (Spero et al., 2015). Although there is no direct evidence demonstrating that quinone oxidoreductase is essential for B. pseudomallei survival under adverse conditions, several reports have demonstrated such a role in other bacteria (Liu et al., 2008; Ryan et al., 2014). For example, in Escherichia coli, significantly higher survival rates were observed in E. coli strain YB overexpressing quinone oxidoreductase than in the control strain when treated with heat shock and oxidative stressors such as H2O2 and menadione (Liu et al., 2008). This supported the hypothesis that quinone oxidoreductase is important for bacterial survival under stress conditions. In our study, we suggested that the down-regulation of quinone oxidoreductase together with the down-regulation of stress response proteins, as mentioned above, most likely led to the decreased survival of the SDO mutant under heat stress conditions.

As in other organisms, during translation in bacterial cells each amino acid is carried by a specific tRNA to the translation site. Prolyl-tRNA synthetase is one of aminoacyl-tRNA synthetases that catalyzes the condensation of a specific amino acid to its cognate tRNA (Crepin et al., 2006). This enzyme is therefore required for bacterial translation, which is a key cellular process. There is no previously reported evidence indicating the role of this enzyme in bacteria survival and virulence, with only one report demonstrating that cysteinyl and lysyl-tRNA synthetases are essential for the growth of Mycobacterium smegmatis (Ravishankar et al., 2016). Further examination of the role of prolyl-tRNA synthetase and its association with SDO is required to further our understanding of their functional significance in B. pseudomallei.

Conclusions

Proteomic analysis revealed a set of B. pseudomallei cellular proteins that were altered in the SDO mutant compared with the wild-type strain. Down-regulation of a set of differentially expressed proteins was detected in the mutant, offering insight into the role of SDO in heat stress tolerance and biofilm formation in B. pseudomallei. In addition, in vitro studies of infected host cells indicated that SDO was also involved in the adhesion of B. pseudomallei to human host cells such as lung epithelial and skin fibroblast cells. The findings of this study provide further insight into the roles and functions of B. pseudomallei SDO, which may be beneficial in the development of prevention and control strategies.

Supplemental Information

Supplemental Information 1 Mass spectrometry for protein identification (Spot 1)

Click here for additional data file.

Supplemental Information 2 Mass spectrometry for protein identification (Spot 2)

Click here for additional data file.

Supplemental Information 3 Mass spectrometry for protein identification (Spot 3)

Click here for additional data file.

Supplemental Information 4 Mass spectrometry for protein identification (Spot 4)

Click here for additional data file.

Supplemental Information 5 Mass spectrometry for protein identification (Spot 5)

Click here for additional data file.

Supplemental Information 6 Growth kinetics of Burkholderia pseudomallei

Click here for additional data file.

Additional Information and Declarations

Competing Interests

Author Contributions

Data Availability

The authors declare there are no competing interests.

Onrapak Reamtong conceived and designed the experiments, performed the experiments, analyzed the data, prepared figures and/or tables, and approved the final draft.

Nitaya Indrawattana, Amporn Rungruengkitkun, Tipparat Thiangtrongjit and Taksaon Duangurai performed the experiments, prepared figures and/or tables, and approved the final draft.

Manas Chongsa-nguan analyzed the data, prepared figures and/or tables, and approved the final draft.

Pornpan Pumirat conceived and designed the experiments, performed the experiments, analyzed the data, prepared figures and/or tables, authored or reviewed drafts of the paper, and approved the final draft.

The following information was supplied regarding data availability:

The raw measurements are available in the Supplementary Files.

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
