# Peer review of "Altered proteome of a Burkholderia pseudomallei mutant defective in short-chain dehydrogenase affects cell adhesion, biofilm formation and heat stress tolerance"

_PeerJ, doi:10.7717/peerj.8659_

## Round 0.1 · original submission · Minor Revisions

The two reviewers agreed that the manuscript is well written but suggested moderate revisions to improve data presentation and to reinforce results with the complemented strain.
Reviewer #01 suggests the inclusion of gene expression data for the SDO complemented strain.
Reviewer #02 makes important observations about data presentation in figures 2 and 3.
Please address carefully all the suggestions by the reviewers in your revised manuscript. Besides, please consider the following when preparing your revision:
1. Include details about bacterial growth phase of the different strains at the moment of protein and RNA extraction, as this can have a significant impact on the protein and RNA expression levels. These details can include OD600nm, CFU counts or even a growth curve for each strain.
2. If available, please include details about differential efficiencies of the primers used for real-time PCR. It is noteworthy that the primer pair targeting 23S rRNA generates an amplicon significantly higher than the average qPCR amplicon and this might impact fold-change calculations. Besides, please include if available some information on the stability of this reference gene under the experimental conditions studied.
3. Please include in Table 2 the number of peptides identified for each protein, besides the sequence coverage found on the Mascot identifications.
4. In Figure 3, please consider performing analysis of variance with a post hoc test, instead of presenting comparisons between pairs with t-test.

Reviewer 1 ·

Basic reporting

The manuscript is relatively well written, the figures are well presented . There are no significant omissions of data.
1. There is a utilisation of Et al. which is not italicised throughout the text and should be corrected.
2. lines 86-87 this sentence should be re drafted as it seems incomplete.
3. Line 92. differential proteins should be replaced with ' differentially expressed proteins'
4. Line 94. the word 'conditions' should be removed.
5. Line 112. dissolved to be replaced with resuspended.
6. Lines 126-127. 'The percentage volume was used for quantification'. Not sure what the authors mean by this?
7. Line 150. Remove the word conventional. Could replace with standard.

Experimental design

The study describe phenotypic assays carried out to assess the results of knocking out BPSS2242, coding for a short-chain dehydrogenase.
Proteomic analysis by Q-TOF MS/MS of the knock out and the wild type indicated 5 differentially abundant proteins. This is a straight forward phenotypic characterisation study so the main question appears to be in establishing if the selected assays adequately demonstrate and support the conclusions.
Figure 1 shows the proteomic profile of a WT and SDO mutant strain. It could be argued that more accurate quantitative approaches such as iTRAQ could have yielded more data but this is not necessarily a cheap technique.
Figure 2. Indicates fold changes in gene expression. This figure is adequate, however, I note that a complement of SDO was made. So why is this not included?
Figure 3. A small set of assays to compare wt, mutant and complement.

Overall, the experiments have been carried out to an acceptable standard , although the expression data from the SDO complement should be shown.

Validity of the findings

The findings are acceptable and not overstated. The conclusions that SDO modulates the expression of several proteins are supported by both proteomic and genetic assays. The authors describe that SDO has been shown to be an important virulence determinant in P. aeruginosa also. However, has any linkage between SDO and regulation of ANY other gene ever been demonstrated in other bacteria?
Also, the finding that expression of the Hsp70 family protein was down-regulated is interesting, but what is the relevance of testing resistance to a temperature as high as 50C, this seems too extreme.

Additional comments

A well written and presented study. The work is robust and the data supports the conclusions.
In my opinion there is just 1 omission that must be further explored. The gene expression data in the SDO complement of the 5 downregulated genes must to be demonstrated.

·

Basic reporting

In their manuscript “Altered proteome of a Burkholderia pseudomallei 1 mutant defective in short-chain dehydrogenase affects cell adhesion, biofilm formation and heat
stress tolerance”, Onrapak Reamtong and co-authors analzed a former generated B. pseudimallei mutant that lacked SDO and showed decreased virulence. Albeit the proteome analysis was carried out with a rather old technique using 2D-PAGE, the results are sound and 5 proteins were found to be significantly altered. The regulation was also confirmed by quantitative PCR. This is carefully executed study and the claims made by the authors are fully supported by the data. The manuscripts also well written. The display of the data (Figure 2&3) could be improved (see specific comments below), but this is only a minor comment. Therefore, I cannot recommend publication if the minor issues below are addressed.

Minor Comments:
Figure 2: The 5 bar charts could be placed in one chart. When the fold change is reported, the first bar with a ratio of 1 does not need to be reported. Also, please consider to show the individual data points in the graph together with the SD. This is a much better presentation of the quantitative data than the current charts.

Figure 3: Similar to above. Please show all individual data points instead of the bar starting at 0, which makes no sense. If there are more than 10 data points per bar, a box plot should be used.

Experimental design

no comment

Validity of the findings

no comment

---

## Round 0.2 · accepted · Accept

The revised version addressed the reviewer's comments in a satisfactory manner.